

# Origins of Mesoscale Mixed Layer Depth Variability in the Southern Ocean

Yu Gao[1,2], Igor Kamenkovich[2], and Natalie Perlin[2]

[1]Scripps Institution of Oceanography, 9500 Gilman Drive, La Jolla, CA, 92093, USA
[2]Rosenstiel School of Marine, Atmospheric and Earth Sciences, University of Miami, 4600 Rickenbacker Causeway, Miami, FL, 33149, USA

**Correspondence:** Yu Gao (yug032@ucsd.edu)

**Abstract.** Mixed-layer depth (MLD) exhibits significant variability, which is important for atmosphere-ocean exchanges of heat and atmospheric gases. Origins of the mesoscale MLD variability at the oceanic mesoscale in the Southern Ocean are studied here in an idealized Regional Ocean-Atmosphere Model (ROAM). The main conclusion from the analysis of the upper-ocean buoyancy budget is that, while the atmospheric forcing and oceanic vertical mixing on average induce the mesoscale variability

of MLD, the three-dimensional oceanic advection of buoyancy counteracts and partially balances these atmosphere-induced vertical processes. The relative importance of advection changes with both season and the average depth of the mixed layer. From January to May, when the mixed layer is shallow, the atmospheric forcing and oceanic mixing are the most important processes, while the advection plays a secondary role. From June to December, when the mixed layer is deep, both atmospheric forcing and oceanic advection are equally important in driving the MLD variability. Importantly, buoyancy advection by ocean

eddies can lead to both local shoaling and deepening of the mixed layer. The role of the atmospheric forcing is then directly addressed by two sensitivity experiments in which the mesoscale variability is removed from the atmosphere-ocean heat and momentum fluxes. The results from these experiments confirm that while the mesoscale MLD variability is controlled by mesoscale atmospheric forcing in summer, the intrinsic oceanic variability and surface forcing are equally important in winter. As a result, MLD variance increases when mesoscale anomalies in atmospheric fluxes are removed in winter and oceanic

advection becomes a dominant player in the buoyancy budget. This study emphasizes the importance of oceanic advection and intrinsic ocean dynamics in driving mesoscale MLD variability, and demonstrates the importance of MLD in modulating the effects of advection in the upper-ocean dynamics.

## 1 Introduction

Mixed layer depth (MLD) changes drastically with time and location in the global ocean. Particularly large changes have been
found in the northern part of the Antarctic Circumpolar Current (de Boyer Montégut, 2004; Dong et al., 2008; Sallée et al., 2010b). This MLD variability can be important for several reasons. The interannual variability of air-sea fluxes of oxygen and carbon dioxide in the Southern Ocean is primarily driven by changes in entrainment of carbon-rich, oxygen-poor waters into the mixed layer during winter convection episodes (Verdy et al., 2007). Gaube et al. (2013) found that eddy-induced Ekman upwelling is at least partially responsible for sustaining positive phytoplankton anomalies in anticyclonic eddies of the South



Indian Ocean, where intensified mixing homogenizes chlorophyll throughout the winter-time mixed layer, which then enables for satellite observations to detect a response to eddy-induced Ekman pumping in anticyclones. Mixed-layer dynamics also modulates the anomalous chlorofluorocarbon uptake by mesoscale eddies in the Drake Passage region (Song et al., 2015), as well as the seasonal variation in the correlation between anomalies of sea level and chlorophyll in the Antarctic Circumpolar Current (ACC) (Song et al., 2018). The oceanic mixed layer is also an important climatic variable, and accuracy of MLD

representation is crucial for closing the mixed-layer heat budget (Dong et al., 2007). MLD can modulate the air-sea thermal coupling by changing the upper ocean heat inertia: shallower (deeper) MLD makes the effective heat capacity lower (higher), and thus the SST is more (less) sensitive to the surface heat flux (Tozuka and Cronin, 2014). As a result, the contribution of surface heat fluxes to surface frontogenesis and frontolysis depends not just on the gradients of the fluxes, but also on the distribution of MLD (Tozuka et al., 2018). Therefore, understanding origins of MLD variability in the Southern Ocean is

important for studies of the oceanic biogeochemical processes, upper-ocean heat budget and air-sea thermal coupling. In this study, we will focus on mechanisms of mesoscale MLD variability and their relations to atmospheric forcing.

Large-scale MLD variability is traditionally attributed to vertical mixing and convection, described by a one-dimensional process (Kraus et al., 1967). Within this framework, several previous observational studies have considered the relative importance of surface buoyancy forcing and wind-induced mixing in the mixed layer formation (Dong et al., 2007, 2008; Sallée et al.,

2010a; Downes et al., 2011; Holte et al., 2012; Sallée et al., 2021). For example, the scaling analysis in Sallée et al. (2010a) shows that the buoyancy forcing in the Southern Ocean dominates the wind mixing by one order of magnitude. In contrast, using two hydrographic surveys and a one-dimensional mixed layer model, Holte et al. (2012) concluded that the wind-driven mixing is central to the winter-time formation of the Subantarctic Mode Water (SAMW), which has been associated with the mixed layer on the equatorward side of the ACC (Rintoul, 2002; Sallée et al., 2006; Holte and Talley, 2009). Holte et al. (2012)

further demonstrated that the mixing driven by buoyancy loss and wind forcing is strong enough to deepen the SAMW mixed layer. This group of studies, however, neglected the role of the oceanic three-dimensional buoyancy advection, which can be an important player in the upper-ocean buoyancy budget and MLD variability. For example, using an eddy-resolving model in the Indo-western Pacific sector of the Southern Ocean, Li et al. (2016) found that in regions with well-defined large-scale jets, there is jet-scale overturning circulation with sinking motion on the equatorward flank and a rising motion on the poleward flank

of the jets. The authors suggested that this overturning is driven by the eddy momentum flux convergence, rather than Ekman pumping or suction, and is thus a result of intrinsic oceanic variability. A follow-up study by Li and Lee (2017) considered a zonal-mean buoyancy balance and found that the downwelling branch of this overturning is responsible for the destratification underneath the mixed layer, and therefore leads to the formation of the deep mixed layer north of the jets. Their study also concluded that the air-sea heat fluxes contribute to the large-scale mixed layer formation along ACC fronts, while the Ekman

advection only accounts for the destratification when the mixed layer initially deepens (Li and Lee, 2017).

Oceanic advection in MLD variability is likely to be more important at the mesoscale than at large scales. In this study, mesoscale anomalies are defined as anomalies on the spatial scales shorter than several oceanic Rossby deformation radii. At these spatial scales, the oceanic currents are particularly strong and highly variable, and can be expected to dominate the upper ocean heat and buoyancy budgets (Tamsitt et al., 2016; Small et al., 2020; Gao et al., 2022). This study will explore the origins



of MLD variability, using an idealized atmosphere-ocean model of a sector in the Southern Ocean. The purpose of the study is to answer the following scientific questions: 1) What is the role of oceanic buoyancy advection in the mixed layer dynamics and MLD variability? 2) What is the role of the surface buoyancy forcing and wind-induced mixing in the mixed layer dynamics and MLD variability? 3) How do the upper-ocean state and MLD variability depend on the mesoscale air-sea coupling?

The first part of the study will analyze the upper ocean buoyancy budget and focus on the relative importance of buoyancy

advection in mesoscale MLD variability. In the second half of this paper, we examine how the mesoscale surface heat and momentum fluxes affect the MLD variability, using two sensitivity experiments with modified atmosphere-ocean coupling. These flux anomalies can influence the mixed layer variability through buoyancy forcing and wind-driven mixing (Kraus et al., 1967; Dong et al., 2007, 2008; Sallée et al., 2010a; Downes et al., 2011; Holte et al., 2012; Sallée et al., 2021), and we will explore the relative importance of these effects. The flux variability can be driven by SST anomalies, because air-sea heat

exchanges act to dampen SST anomalies: (Putrasahan et al., 2013; Gao et al., 2022). Additionally, acceleration (deceleration) of surface winds over warm (cold) SST anomalies modulate the turbulent air-sea heat exchange (Xie, 2004; Small et al., 2008), which may be important when winds are strong. These fluxes can also result from internal atmospheric dynamics, but we will not attempt to study the intrinsic and SST-forced sources of flux variability separately.

## 2 Methodology

### 2.1 Regional Ocean–Atmosphere Model and Sensitivity Experiments

Regional Ocean–Atmosphere Model (ROAM) is a semi-idealized high-resolution regional coupled model. Here we provide a brief description of the model, since it is described in greater detail Gao et al. (2022) and Perlin et al. (2020). The atmospheric component is the US Navy Ocean-Atmosphere Mesoscale Prediction System (Hodur, 1997), forced by lateral boundary conditions from the global reanalysis dataset. The global reanalysis dataset is derived from 6-hourly 0.25°global NCEP FNL

analysis and Global Forecast System (GFS). The atmospheric component has two nested domains: the inner domain that is fully coupled with the ocean component, and the outer domain that is one-way coupled with the observed SST. The inner and outer atmospheric domains have 9- and 27-km horizontal grid-box sizes, respectively; both domains have 49 vertical layers. The transition between the inner and outer domains uses a blending scheme that ensures a gradual transition between ROMS-simulated SST and observed SST surrounding the ocean grid. The ocean component is the Regional Ocean Modeling System

(ROMS) (Shchepetkin and McWilliams, 2005), formulated in a zonal re-entrant channel with north and south sponge boundaries that help to keep the meridional density gradient close to a desired value. The atmospheric forcing at sea surface comes from the atmospheric component. The horizontal ocean grid box is 2.5 km and there are 30 sigma layers in the vertical. The channel is 2800 km long and 1120 km wide. The large-scale stratification in the ocean is chosen from a region in the Western Indian sector of the Southern Ocean. The domain and bathymetry are shown in Figure.1 in Gao et al. (2022). The simulated

large-scale SST, surface salinity and MLD are in general agreement with observed values (Fig.1a-c), and exhibit a primarily eastward flow with an intense jet in the northern part of the domain (Fig.1d). Mesoscale anomalies are clearly visible in all fields (Fig.1).





In addition to the control simulation, we carried out two sensitivity experiments: the Smooth-Fluxes experiment and the Smooth-Winds experiment, in which the mesoscale air-sea heat fluxes (turbulent and radiative) and wind stress respectively

are removed with spatial smoothing by a 300km-by-300km running boxcar average.

The Control and sensitivity simulations are spun up for three years, following a 12-year spin-up of the uncoupled ocean model. The length of the spin-up is deemed to be sufficient to reach equilibrium for the upper ocean temperature, based on the equilibrium of the domain-average Eddy Kinetic Energy. The Control and the sensitivity experiments are then ran for two more years, using the atmospheric forcing from July 2015 to June 2017. Although the duration of the simulations is relatively

short, the main analysis is based on relations within numerous mesoscale features, which will ensure that the conclusions are statistically significant.

### 2.1.1 Ocean Mixed Layer Buoyancy Budget

We diagnose spatial and temporal variations in the upper-ocean buoyancy budget here in order to examine the origins of the MLD variabiltiy. Following Li and Lee Li and Lee (2017), the depth-dependent buoyancy budget can be expressed as

$$
\underbrace{\frac{\partial b}{\partial t}}_{\text{buoyancy tendency}} = \underbrace{-\frac{g}{\rho_0}[\alpha_\theta(-\mathbf{u}\cdot\nabla\theta)+\beta_S(-\mathbf{u}\cdot\nabla S)]}_{\text{buoyancy advection}} + \underbrace{-\frac{g}{\rho_0}[\alpha_\theta D(\theta)+\beta_S D(S)]}_{\text{diffusion}} + \underbrace{B}_{\text{buoyancy forcing}}
\tag{1}
$$

where $b \equiv -g\frac{\delta\sigma}{\rho_0}$ is buoyancy, $\delta\sigma$ is the potential density anomaly, $\rho_0 = 1027 kgm^{-3}$ is the reference density, $\mathbf{u}$ is the three-dimensional velocity vector, $\alpha_\theta$ is the thermal expansion coefficient, $\beta_S$ is the haline contraction coefficient, $\theta$ is the potential temperature, $S$ is the salinity, $D(\theta)$ is the potential temperature diffusion, $D(S)$ is the salinity diffusion, and the buoyancy forcing term $B$ is

$$
B = \frac{-g\alpha_\theta}{\rho_0 C_p}\frac{\partial q}{\partial z}
\tag{2}
$$

where $C_p = 3850 J/(kgC)$ is the specific heat of seawater, $q$ is the downward shortwave flux (units $Wm^{-2}$). At the ocean surface, the downward surface heat fluxes can be decomposed into four components:

$$
q|_{z=0} = Q_{net} = Q_{SW} + Q_{LW} + Q_s + Q_l
\tag{3}
$$

where $Q_{SW}$ is the shortwave heat flux, $Q_{LW}$ is the longwave heat flux, $Q_s$ is the turbulent sensible heat flux and $Q_l$ is the

latent heat flux.

In order to examine the evolution of the vertical stratification in the upper ocean, we take the vertical derivative ($\frac{\partial}{\partial z}$) of the buoyancy and compute the buoyancy frequency squared ($N^2$):

$$
N^2 = \frac{\partial b}{\partial z} = \frac{-g}{\rho_0}\frac{\partial\sigma}{\partial z}.
\tag{4}
$$



As a measure of stratification, $N^2$ is positive, corresponding to a stable water column. Taking the vertical derivative of Eq.1
and combining with Eq.4, we can get the depth-dependent buoyancy frequency budget below (Li and Lee, 2017):

$$\underbrace{\frac{\partial N^2}{\partial t}}_{\text{tendency}} = \underbrace{\frac{\partial}{\partial z}(-\mathbf{u} \cdot \nabla b)}_{\text{advective restratification}} + \text{residual}. \tag{5}$$

The first term on the right-hand side quantifies the re-stratifying effect of advection on the water column. Its positive values
indicate that advection increases the stratification and hydrostatic stability. In order to emphasize net effect of all processes on
the MLD, we integrate Eq.5 in the vertical to get the buoyancy frequency budget over the mixed layer:

$$\underbrace{\int_{-H}^{0} \frac{\partial N^2}{\partial t}}_{\text{mixed layer integrated tendency}} = \underbrace{(-\mathbf{u} \cdot \nabla b)\Big|_{z=0} - (-\mathbf{u} \cdot \nabla b)\Big|_{z=-H}}_{\text{advective re-stratification of ML}} + \text{residual} \tag{6}$$

where H is the MLD, defined here as the depth at which the potential density increases by 0.03 $kg\,m^{-3}$ relative to the surface
value. The mixed layer integrated tendency term indicates whether the mixed layer as a whole is becoming more or less
stratified. The advective term is now in the form of the difference between surface buoyancy advection and advection at the
base of the mixed layer. Its positive values indicate that the advection acts to shoal the mixed layer. If the buoyancy advection
increases the buoyancy difference between the surface and the base of the ML, the term has a re-stratifying tendency and mixed
layer shoals. If the buoyancy advection decreases the buoyancy difference between the surface and the base of the mixed layer,
the advection term has a destratifying tendency and mixed layer deepens.

The residual term represents contribution of the atmospheric buoyancy forcing and oceanic subgrid mixing. These processes
are related, since surface heat exchanges are associated with changes in stratification, vertical mixing and convection, whereas
stronger momentum fluxes lead to intensification of mixing. The atmospheric forcing is, thus, implicit in the vertical mixing
term and its direct role in the budget is difficult to calculate. We will not attempt to do it in our analysis. The residual term also
includes diffusion and cabelling terms due to nonlinear effects Li and Lee (2017). These buoyancy budget terms are calculated
from daily snapshots of the model output, in order to minimize sampling errors.

## 3   Results

We begin by exploring the one-dimensional view on the mixed-layer deepening and shoaling, which implies that variations in
MLD are primarily caused by changes in surface density. The large-scale MLD is indeed well correlated with the SST (correla-
tion coefficent is -0.7), with cooler (warmer) SST leading to deeper (shallow) MLD. In contrast, the relationship becomes more
complicated for mesoscale anomalies, which are defined as departures from the 300km-by-300km running boxcar average.
The mesoscale MLD and SST anomalies are only weakly correlated, with the average correlation coefficient of -0.2 (Fig.6).



The relatively weak correlation can be explained by the importance of oceanic advection in MLD variability, which will be explored in the following sections.

### 3.1    Mixed Layer Buoyancy Balance

We first examine the monthly depth-dependent buoyancy budget at the depth-latitude cross section taken at X = 2000 km in the zonal direction, in order to identify the key processes that contribute to the seasonal variability in MLD. We also examined

section X = 1000km and reached the same conclusion. The evolution of the upper-ocean stratification and hydrostatic stability is measured by the tendency in the buoyancy frequency squared ($N^2$), see eq.5. From January to March, the positive $N^2$ tendency (re-stratifying signal) at the base of the mixed layer indicates that the water column is getting more stable during the warm summer months (Fig.2a). The mixing and atmospheric forcing ("residual") term makes the largest contribution to the $N^2$ tendency everywhere within the mixed layer and is spatially correlated with the tendency (Fig.2b). The buoyancy advection

has a re-stratifying effect over most of the mixed layer, but is not spatially correlated with the $N^2$ tendency (Fig.2c).

     Starting in April, the $N^2$ tendency turns negative at the mixed-layer base, which indicates that the water column is getting less stable and the mixed layer begins to deepen (Fig.2d). From May to August, the $N^2$ tendency is negative around the mixed layer base, which indicates that the mixed layer continues to deepen, consistent with the destratifying effects of the austral winter-time surface cooling (Fig.2g). The destratifying effects by the mixing and atmospheric forcing explain most of the $N^2$

tendency around the base of the mixed layer (Fig.2e, h). The advective restratification term stays positive, which implies that the buoyancy advection acts to shoal the mixed layer (Fig. 2f, i).

     In September, the $N^2$ tendency is nearly zero while the mixed layer is the deepest (Fig.3a). The mixing and atmospheric forcing ("residual") term (Fig.3b) continues to destratify the mixed layer, but these effects are now very small everywhere, except near the surface, where they are compensated by the buoyancy advection (Fig.3c). The advective term is also very small

in most of the domain (Fig.3c). In December, the $N^2$ tendency turns mostly positive at the base of the mixed layer (Fig.3g), meaning that the water column is re-stratifying, which leads to the shoaling of the mixed layer. In December, the destratifying effect of the residual term in the upper 80 meter is compensated by the restratification (Fig.3h-i), and the tendency is very weak within the mixed layer.

     We next present the depth-integrated monthly mixed layer buoyancy budget in eq.6. The advantage of this analysis is that it

allows us to focus on the lateral mesoscale structure in all budget terms. From January to March (austral summer), when the MLD is relatively shallow, the evolution of the vertical density contrast (integrated $N^2$ tendency) is caused by the atmospheric forcing and ocean mixing (Fig.4a-b). The advection term is much smaller than the residual term, except in the jet region (Fig.2), where it exhibits small-scale anomalies with the spatial scale of O[10km] (Fig.4c). These anomalies are persistent throughout the year in the jet region. Note that despite the importance of short spatial scales, the dynamics everywhere in the domain is still

geostrophic, since we verified that the Rossby number, measured by a ratio between relative and planetary vorticity, is small. In June, when the mixed layer is deepening, the advection and residual terms are both large and tend to balance each other in the southern part of the domain (Fig.4e-f). In September, when the MLD reaches its maximum, the advection and residual





terms balance each other and the tendency term is small (Fig.4g-i). From September to December, both advection and residual continue to play equally important roles in the buoyancy budget.

In summary, both buoyancy advection and residual processes (mixing and atmospheric forcing) play important roles in MLD variations. There is, however, significant seasonality in the buoyancy budget. The mixing and atmospheric forcing are the dominant terms in summer and autumn (January to May), when they drive variations in MLD, while the buoyancy advection becomes equally important in winter and spring (from June to December). In late winter, the advection balances the mixing and atmospheric forcing. The MLD modulates the relative importance of advection, with the thicker and more thermally inert

mixed layer being less controlled by the atmospheric forcing. Another important result is that buoyancy advection by ocean eddies can have both re-stratifying and destratifying effects, whereas some previous studies only account for their overall re-stratifying effect (Fox-Kemper et al., 2008; Fox-Kemper and Ferrari, 2008).

### 3.2    Mixed-Layer Response to Mesoscale Atmospheric Forcing

The results so far have demonstrated the importance of both atmospheric forcing and oceanic buoyancy advection in driving

mesoscale MLD variability. The conclusions were, however, drawn from spatial relations between the corresponding terms in the buoyancy budget, which is not sufficient for quantifying contributions from each process in such a highly nonlinear system. For example, in addition to directly affecting stability of the upper ocean, the mesoscale atmospheric forcing can also influence buoyancy advection and oceanic mixing. The sensitivity experiments described in this section will directly inquire about the role of atmospheric heat and momentum fluxes in the corresponding dynamics.

Downward heat fluxes and wind stress at sea surface in the Control experiment exhibit well-pronounced mesoscale variability (Fig.1e-f). As is discussed in the Introduction, this variability is both a result of internal atmospheric variability and a response to mesoscale SST anomalies. In the Smooth-Fluxes and Smooth-Winds experiments, we remove these mesoscale anomalies in the air-sea heat fluxes and wind stresses respectively using a 300km-by-300km moving average. This is done during coupling, so the ocean model is forced by either the large-scale heat fluxes or wind stresses.

The zonal-mean structure of the surface heat flux and wind stress in the Control, Smooth-Fluxes and Smooth-Winds experiments are very close to each other (within 10% or less, not shown). Mesoscale heat flux anomalies are particularly large in the jet region in the northern part of the domain (not shown). In the Control experiment, the turbulent heat fluxes are negatively correlated with SST anomalies, and this negative relationship is stronger in the northern domain where the ocean currents themselves are stronger (Fig.5a). This negative correlation indicates that the mesoscale air-sea heat fluxes dampen the SST

anomalies. As a result of smoothing out the mesoscale heat fluxes, the relationship between SST anomalies and turbulent heat fluxes is significantly weaker in the Smooth-Fluxes experiment, indicating a reduced atmospheric feedback on SST anomalies (Fig.5a).

In the Smooth-Winds experiment, the ocean does not "feel" the mesoscale wind stress anomalies, which are mainly a response to SST variability (Perlin et al., 2020). Specifically, mesoscale wind stress is positively correlated with SST anomalies

in the control experiment, and this relationship is stronger in the northern part of the domain where the ocean currents are stronger (Fig.5b). Positive correlation indicates that the wind stress is stronger (weaker) over warm (cold) SST anomalies.



Wind stress variability, in turn, induces anomalous Ekman circulation, and anomalies in wind speed can modulate air-sea heat exchanges. As a result of smoothed-out mesoscale wind stress anomalies, the relationship between SST anomalies and wind stress is significantly weaker in the Smooth-Winds experiment (Fig.5b). The effects on the SST variability itself are, however, 215 weaker than in the Smooth-Fluxes case.

As a consequence of the smoothing of atmospheric forcing, there are significant differences in mesoscale SST variability between the Control and sensitivity experiments. Atmosphere feeds back on mesoscale SST anomalies via turbulent air-sea fluxes and wind stress. Smoothing of these fluxes weakens this feedback. For example, the lack of atmospheric thermal damping in the Smooth-Fluxes case leads to the enhanced mesoscale SST variability from January to May, as shown in the domain-average 220 Root Mean Squared (R.M.S.) SST anomalies (Fig.7). Note also that a part of mesoscale variability in surface heat/momentum fluxes is not an SST-forced response, but is instead a result of intrinsic atmospheric variability.

The R.M.S. mesoscale MLD anomalies exhibit noticeable differences (up to 20%) between the Control run and the sensitivity experiments (Fig.8). Fig.8b shows the ratio between the R.M.S. of MLD between the simulations. From January to May, the results suggest that mesoscale air-sea heat fluxes enhance MLD variability, because it decreases when the mesoscale surface 225 fluxes are filtered out. During this time period, the MLD is shallow and its variability is mainly driven by the atmospheric forcing and ocean vertical mixing (Fig.4), and it is not surprising that the reduction in mesoscale surface heat and momentum fluxes leads to weaker MLD variations. From September to November (local spring), the opposite is true: the mesoscale MLD variability is weaker in the Control case than in the sensitivity runs. This result may look counter-intuitive and deserves further discussion. During the local spring in the Control experiment, the buoyancy advection is balanced by the mixing and 230 atmospheric forcing in most of the domain (Fig.4). This balance is characteristic of the thick mixed layer with its high inertia. When one of the two main terms in this balance, the mesoscale surface flux, is artificially reduced in the sensitivity runs, the balance is no longer valid and the MLD variability is enhanced.As briefly discussed in Section 2, the significance of the changes is guaranteed by a large number of mesoscale anomalies. In order to further test the robustness of the aforementioned conclusions, we calculated the same R.M.S. MLD anomalies in the western half, eastern half, and the northern half (which 235 contains the jet region) subsets. We found qualitatively the same seasonality in all three subsets (not shown).

Compared to the mesoscale variability in MLD, the mesoscale atmospheric forcing makes a small difference (below 6%) in the domain-average MLD (Fig.9). The spatial distribution of the annual-mean MLD differences exhibits a dipolar structure with larger magnitude in the northern part of the jet region and smaller values in the southern part (not shown), which is similar to the annual-mean difference in the ocean current speed. The shape of the patterns and their similarity indicate a southward 240 shift and suggest a relation between the jet and the meridional slope in the MLD. This conclusion is confirmed by the vertical cross sections. The magnitudes of the changes are, however, small and their significance is hard to evaluate.

## 4 Conclusions

The mixed layer in the Southern Ocean modulates ocean-atmosphere exchanges of heat and atmospheric gases, by changing the effective upper-ocean thermal inertia. In this study, we explore the origins of MLD variability using both buoyancy budget



analysis in ROAM simulations and sensitivity experiments. The results demonstrate that oceanic advection plays a central role in the buoyancy budget and that this role is amplified when the mixed layer is deep.

    In the budget analysis, the time evolution of the buoyancy frequency $N^2$ represents the re- or destratifying tendency in the water column. This tendency is driven by the two main processes: the three-dimensional advection of buoyancy and the atmospheric forcing and upper-ocean mixing (the "residual term"). The mixing and surface fluxes of heat and momentum in
the residual term are closely intertwined and represent the one-dimensional vertical forcing. The budget also exhibits strong seasonal variations due to the changes in mixed-layer heat inertia. In summer and autumn, when the mixed layer is shallow, atmospheric forcing and ocean mixing dominate the budget while the buoyancy advection is of minor importance and its significance is mainly restricted to the jet region. In winter and spring, when the mixed layer is deep and its inertia is large, the buoyancy advection becomes more important and balances the mixing and atmospheric forcing. Another important result
is that the advection can have both re-stratifying (mixed-layer shoaling) and destratifying (mixed-layer deepening) effects. In fact, the destratifying advective effects are widespread at mesoscale, which challenges a common view that mesoscale eddies always restratify the upper ocean.

    The importance of mesoscale surface heat/momentum fluxes is directly addressed in two sensitivity experiments, the Smooth-Fluxes and Smooth-Winds experiments. In the Smooth-Fluxes experiment, the ocean component is forced with large-scale
surface heat fluxes, whereas the atmospheric component still reacts to the full SST anomalies. Similarly, the ocean component in the Smooth-Winds experiment is forced by large-scale surface wind stress. The overall effect of mesoscale heat-flux forcing is more significant than the mesoscale wind-stress forcing. The significance is, however, limited to mesoscale variability, whereas the large-scale MLD is largely insensitive to the presence of mesoscale atmospheric forcing. For the mesoscale MLD variability, there is seasonal variation in the importance of mesoscale atmospheric forcing, which is consistent with our
conclusions from the budget analysis. The atmospheric forcing dominates the summer-time mesoscale buoyancy budget in the Control simulation, and the absence of mesoscale heat and momentum fluxes leads to smaller MLD anomalies in the sensitivity runs. In winter, there is a balance between the buoyancy advection, mixing and atmospheric forcing in the Control simulation and variations in water-column stability are small. The reduction in the variability of the atmospheric forcing in the sensitivity experiments breaks this balance and leads to the enhanced advection-driven variability in MLD.

This study explores the regime of mesoscale atmosphere-ocean coupling, which is important for the upper-ocean dynamics and mixed layer variability. The mixed layer variability has implications for both physical and biochemical processes in the Southern Ocean. The findings demonstrate the tendency of atmospheric forcing and advection to compensate each other in the mixed layer buoyancy budget, which is especially pronounced in local summer. This result implies that neglecting or underestimating the role of oceanic advection would lead to biases in mixed layer variability, upper ocean dynamics, and air-
sea exchanges of heat and atmospheric gases. Conclusions made from this semi-idealized model have potential caveats. For example, in contrast to mesoscale MLD anomalies, large-scale MLD are largely insensitive to mesoscale atmospheric forcing. This lack of sensitivity in large-scale MLD is likely to be partially explained by the use of the sponge boundaries that keep the large-scale stratification from changing. The ocean component is an idealized model with generic mesoscale topography, which means it lacks many realistic features of the Southern Ocean. Specifically, the idealized topography cannot represent



ridges and plateaus of the Southern Ocean, and the atmospheric forcing does not represent the full range of the atmospheric conditions in the region. In addition, although we believe that this idealized modeling study successfully describes the main processes involved in the MLD variability, quantitative conclusions can be different in parts of the real Southern Ocean.

*Code and data availability.* The original dataset of the numerical simulations used in this study is too large to directly archive or transfer. Instead, we provide the aggregated data and the information necessary to replicate the results and analyze the data. The aggregated data and

the Python code is shared in this data repository: https://doi.org/10.17604/0bkf-p943.

*Author contributions.* Yu Gao: Conceptualization, Methodology, Software, Formal analysis, Investigation, Data Curation, Writing - original draft, Writing - review and editing, Visualization. Igor Kamenkovich: Conceptualization, Methodology, Software, Formal analysis, Investigation, Data Curation, Writing- Original draft, Writi ng - review and editing, Visualization, Supervision, Project administration, Funding acquisition. Natalie Perlin: Software, Investigation, Data Curation, Writing - review and editing, Funding acquisition.

*Competing interests.* The authors declare that they have no known competing financial interests or personal relationships that could have appeared to influence the work reported in this paper.

*Acknowledgements.* This study is supported by the National Science Foundation (NSF) Research, USA, Award No. 1559151 and 1849990, and The National Aeronautics and Space Administration (NASA) grant No. 80NSSC20K1136. We acknowledge the computing resources provided by the University of Miami's Center of Computational Science and the high-performance computing support from Cheyenne

provided by NCAR's Computational and Information Systems Laboratory, sponsored by NSF. We thank Dr. Michael Spall of Woods Hole Oceanographic Institution and Dr. Lisa Beal of University of Miami for their helpful comments and suggestions. We also thank the editors and the anonymous reviewers for their thoughtful comments.



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





**Figure 1.** Examples of model simulated fields (values on on September 1st, 2016): a) sea surface temperature, b) sea surface salinity, c) mixed layer depth, d) surface ocean current speed, e) downward sea surface heat fluxes, and f) surface wind stress.



**Figure 2.** Monthly-mean depth-dependent buoyancy budget in the Control case at the vertical cross section X = 2000 km in January, April and July, respectively. a)-c)$N^2$ tendency (shading, unit: $s^{-3}$) and MLD (white line, unit: meters), d)-f) residual term (shading, unit: $s^{-3}$) and MLD and g)-i) buoyancy transport contribution term (shading, unit: $s^{-3}$) and MLD.





**Figure 3.** Monthly-mean depth-dependent buoyancy budget in the Control case at the vertical cross section X = 2000 km in September, November and December, respectively. (a)-(c) $N^2$ tendency (shading, unit: $s^{-3}$) and MLD (white line, unit: meters), (d)-(f) residual (shading, unit: $s^{-3}$) term and MLD and (g)-(i) buoyancy transport contribution term (shading, unit: $s^{-3}$) and MLD.







**Figure 4.** Monthly-mean mixed layer-integrated buoyancy budget in the Control case in January and March, 2016: (a)-(c) $N^2$ tendency, (d)-(f) residual term and (g)-(i) buoyancy transport contribution term in Eq.6 (unit: $m/s^{-3}$).



**Figure 5.** (a) Zonally averaged correlation coefficient between mesoscale SST anomalies and downward sea surface heat-flux anomalies; (b) zonally averaged correlation coefficient between SST anomalies and sea surface momentum stress. The Control case is shown by the green line and the Smooth-Fluxes (Smooth-Winds) case by the red (blue) line.





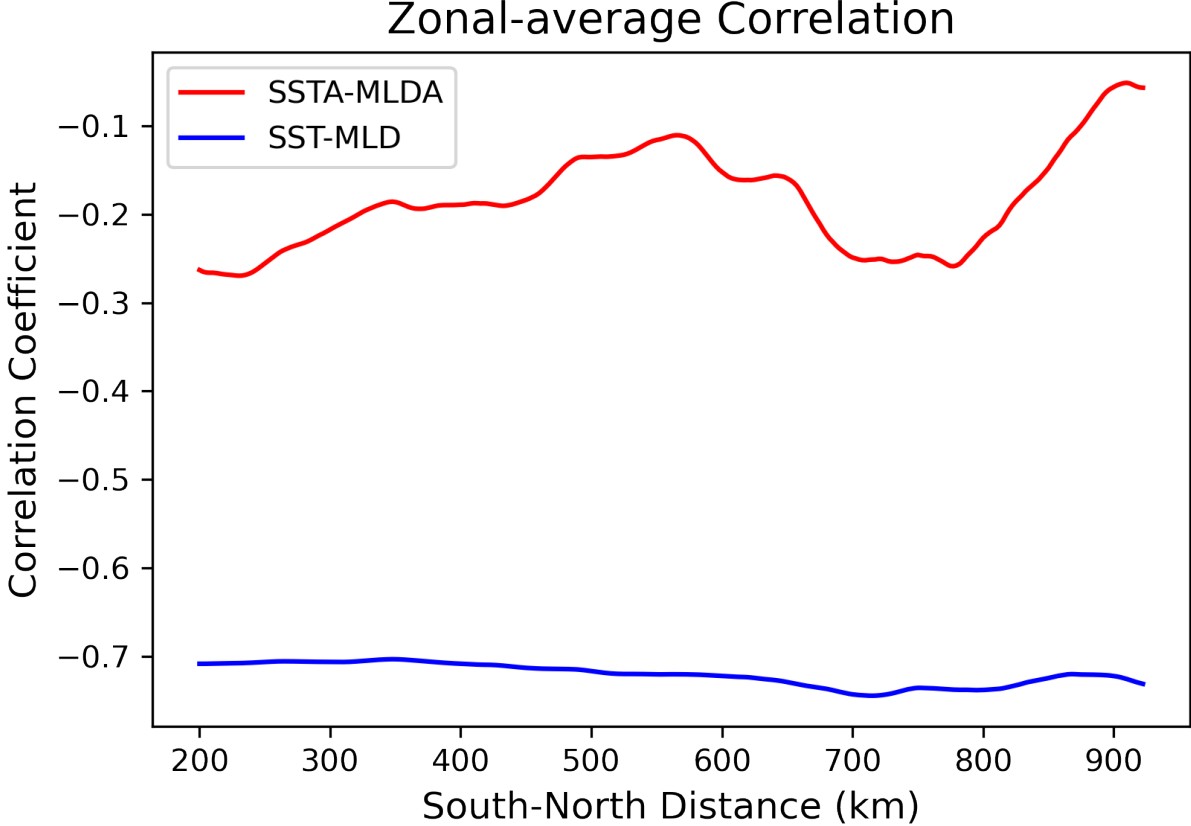

**Figure 6.** Zonally averaged correlation coefficient between SST and MLD (blue line) and zonally averaged correlation coefficient between SST and MLD anomalies (red line). The anomalies are defined as the departure from the the 300km-by-300km running boxcar average.







**Figure 7.** (a) Time series of domain averaged Root Mean Squared (r.m.s.) SST anomalies in the control (green), Smooth-Fluxes (red) and Smooth-Winds (blue) experiment; (b) Time series of the domain averaged differences in SST r.m.s between the control run and Smooth-Fluxes experiment (red line) and between the control and Smooth-Winds experiment (blue line).



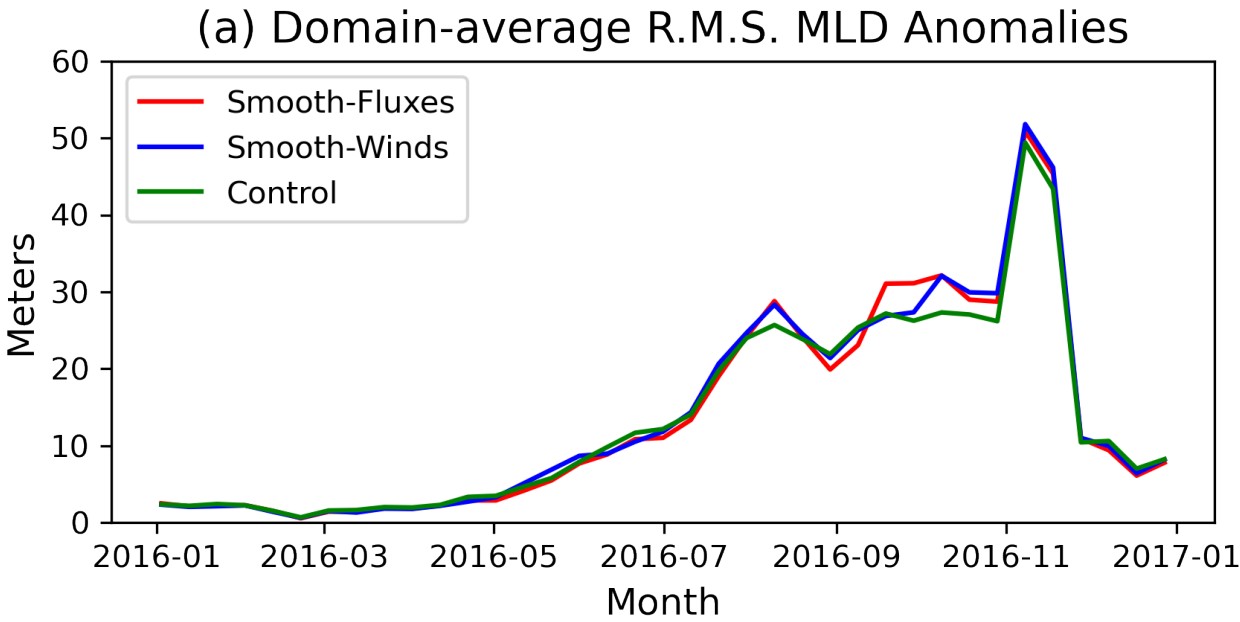

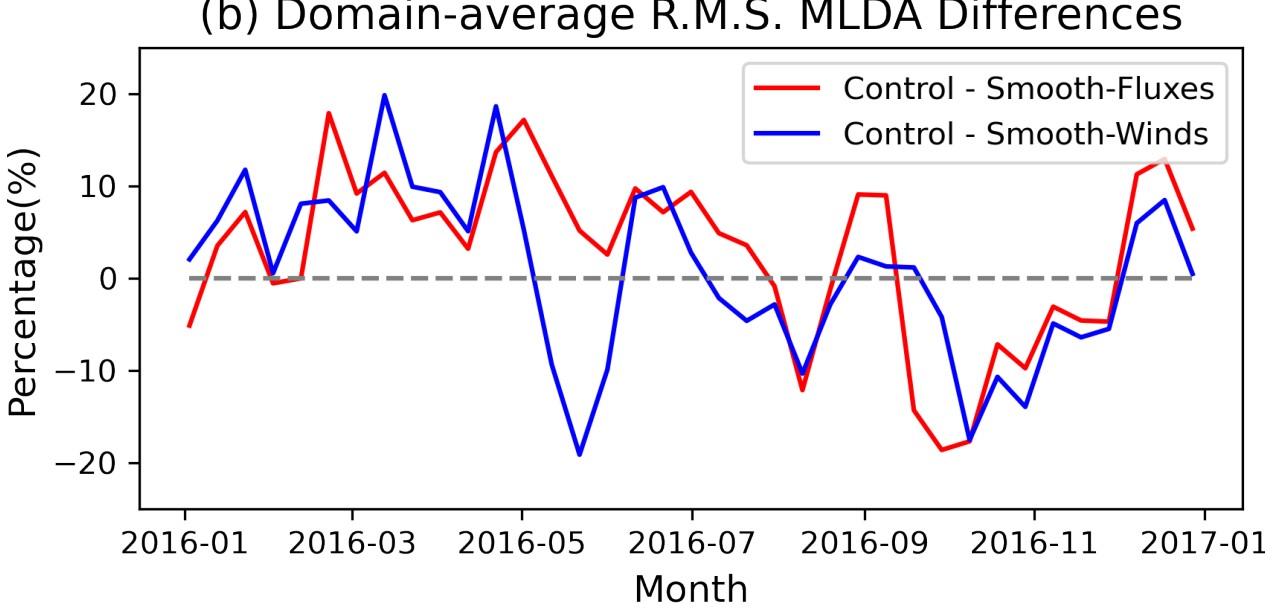

**Figure 8.** (a) Time series of domain-average Root Mean Squared (r.m.s.) MLD anomalies in the control (green), Smooth-Fluxes (red) and Smooth-Winds (blue) experiment; (b) Time series of the domain averaged r.m.s. MLD anomalies differences (in percentage) between the control run and Smooth-Fluxes experiment (red line) and between the control run and Smooth-Winds experiment (blue line).







**Figure 9.** (a) Time series of the domain averaged MLD in the control (green), Smooth-Fluxes (red) and Smooth-Winds (blue) experiments; (b) time series of the domain averaged MLD differences between the control and Smooth-Fluxes experiment shown by the red line and the difference between the control and Smooth-Winds experiments shown by the blue line. (unit: meters)