# Peer review of "Origins of Mesoscale Mixed Layer Depth Variability in the Southern Ocean"

_EGUsphere, 2022_

## Referee Comment (RC2)

**Review: "Origins of Mesoscale Mixed Layer Depth Variability in the Southern Ocean"**

Authors: Yu Gao, Igor Kamenkovich & Natalie Perlin

**Summary**

This study, using a semi-idealized high-resolution Regional Ocean-Atmosphere Model (ROAM), explores the role of mesoscale atmosphere-ocean coupling in the upper-ocean dynamics and mixed layer variability. The authors conduct two sensitivity experiments: *Smooth-Fluxes* and *Smooth-Winds*. The results from these experiments show the distinct contributions of ocean buoyancy advection, mixing and atmospheric forcing to the MLD variability in summer and winter. Overall, the effect of mesoscale heat-flux forcing is more significant than the mesoscale wind-stress forcing.

The paper contains a complete science narrative and interesting results, which are convincingly supported by a thorough study of mixed-layer buoyancy budget. The paper is well-written, but could be made a bit clearer in some specific places, as I listed below. I recommend publication after a minor revision.

**Minor comments**

- **Description of model domain.** Lines 91–92: "*Mesoscale anomalies are clearly visible in all fields (Fig.1).*" Because this paper is focused on the ocean mesoscale dynamics, it is worth to add a snapshot of Eddy Kinetic Energy (EKE). I suggest replacing the SSS pattern (Fig. 1b) with an EKE pattern.

- **MLD pattern in model domain.** In the model domain, which is a region in the Western Indian sector of the Southern Ocean, the deep MLD forms in the southern and eastern part of the domain (Fig. 1c). In fact, this deep MLD distribution is quite different from that in most other regions of the Southern Ocean, where the deep MLD forms on the northen flank of the ACC jets. More discussion/clarification is needed here.

- Figure 6. Line 144: I am confused about the logic here. Should the authors show the correlation between SSTA and **MLD**, instead of SSTA and MLDA, in Fig. 6? In this way,

the authors can contrast the contributions of SST with and without mesoscale anomalies to the MLD variability.

- Figure 8. Can the authors comment on why the RMS MLD anomalies in the two sensitivity experiments show a similar response in time?

- Figure 9. The signals of two sensitivity experiments in the domain average are small and not very clear. I suggest to conduct the same calculation, but only averaged for the areas where the wintertime MLDs in the control experiment are deeper than a certain threshold, i.e. $\geq$400 m. I expect that the signals would become clearer.

- Figure 3. There are some 'white spots' close to the surface in Fig. 3e,f. This issue can be solved by modifying the colorbar.

---

## Author Response (AR1)

**Author's Response**

**Referee #1**

Major comments:
1) Although not considered in this manuscript, the stratification below the mixed layer can also influence MLD variability. This effect could also be discussed.

- Thank you for your thoughtful comments on our paper. We appreciate your feedback on our discussion of the monthly depth-dependent buoyancy budget at the depth-latitude cross section taken at X = 2000 km in the zonal direction. We agree that considering the stratification below the mixed layer is crucial for identifying the key processes that contribute to the seasonal variability in MLD. As you noted, our analysis indicates that from January to March, the re-stratifying signal at the base of the mixed layer indicates that the water column is becoming more stable during the warm months (see lines 155-165).

2) Although the authors did not distinguish them, **cyclonic and anticyclonic eddies** may contribute differently. Not only do they have different impacts on oceanic stratification, which is pointed out in the above comment, they have different impacts on the overlying atmosphere. The authors may find a paper by Gaube et al. (2015) useful.
Gaube et al. (2015) Satellite Observations of Mesoscale Eddy-Induced Ekman Pumping. J. Phys. Oceanogr., 45, 104–132.

- Here we use "eddy" and "mesoscale" interchangeably, but here "eddies" doesn't solely mean coherent vortices. We agree that it is important to clarify that in our analysis, "eddy" refers not only to coherent vortices but also to all mesoscale processes that the filter can smooth out. Regarding the contribution of cyclonic and anticyclonic eddies to Ekman upwelling and downwelling, respectively, we appreciate the reference to Gaube et al. (2015), which highlights the importance of distinguishing between these two types of eddies. However, our focus in this study is on the statistical characteristics and parameterization of ocean eddies, rather than on the separation of cyclonic and anticyclonic eddies.

3) Section 3.1: I think buoyancy budget analyses alone cannot isolate effects of ocean eddies. **Ekman currents** can also contribute to the buoyancy advection term. Thus, at this stage, the authors have not shown that "buoyancy advection by ocean eddies can have both re-stratifying and destratifying effects, whereas some previous studies only account for their overall re-stratifying effect (Lines 185-187). I recommend the authors to support this statement with more evidence.

- Thank you for your comments on the role of Ekman advection in our study. While we did not attempt to separate Ekman advection from other advection processes, we agree that it is an important consideration in understanding the dynamics of oceanic eddies.

The advection term in our analysis is composed of two parts: the locally forced Ekman advection driven by the wind, and the geostrophic and ageostrophic processes that exhibit strong mesoscale variability. We expect the ageostrophic variability to be much weaker than the geostrophic part since the Rossby numbers in both the SC and WC domain are much smaller than 1.

The Ekman current, as part of the advection, is expected to play a role in regions with strong wind variability and SST gradients. While we did not specifically discuss the role of Ekman heat advection in our study, we acknowledge its role in the conclusion section (lines 251 - 256).

4) The 300 km x 300 km box-car averaging not only smooth out meso-scale eddy variability, but also frontal-scale variability.

- Thank you for your comment regarding mesoscale variability and our use of the terms "eddy" and "mesoscale" interchangeably. We agree that mesoscale variability includes frontal-scale variability, and that the term "eddy" encompasses not only coherent vortices but also all mesoscale processes that can be smoothed out by the filter.

To clarify, in our study we use the term "eddy" to refer to all mesoscale processes, including frontal-scale variability, which are filtered out in our analysis. We appreciate your input and will make sure to clarify this terminology in the revised manuscript at lines 146 - 148.

5) Smooth-Winds experiment: The statement "anomalies in wind speed can modulate air-sea heat exchanges" in Lines 212-213 made me wonder whether turbulent heat fluxes that are a function of wind speed are also modified in the Smooth-Winds experiment. If so, this experiment cannot isolate effects of mesoscale wind stress.

- We agree that turbulent heat fluxes are a function of wind speed, but in our Smooth-Winds experiment, we specifically focused on the effect of mesoscale wind stress by smoothing wind stress instead of wind speed.

To clarify the naming of our experiments, we agree that using "Smooth-heat-flux" and "Smooth-momentum-flux" instead of "Smooth-fluxes" and "Smooth-winds" would more accurately reflect the focus of our analysis. Here we can use an example of an oil spill on the sea surface as a useful analogy to explain the concept of decorrelation between wind stress and wind speed. When oil spill happens on sea surface, wind speed stays the same while wind stress became very low because of low drag coefficient. This is how decor-relation happens between wind stress and wind speed. Even if wind speed remains constant, a change in the drag coefficient can result in a decrease in wind stress and therefore affect turbulent heat fluxes.

6) In Figs. 7-9, the authors are showing time-series for a single year, but because of large-amplitude high-frequency variability, it may obscure seasonal variations in some cases.

- We agree that the high-frequency variability shown in Figs. 7-9 may make it difficult to discern seasonal variations in some months. However, our goal in showing these time-series was to illustrate the variability and trends in the data over the entire time period. Most of the seasonal variations happens in fall/winter when mixed layer deepens.

Minor comments:
1) Lines 2-3: In this sentence, "mesoscale" appears twice and seems redundant. I recommend removing one of them.

- We removed "at the oceanic mesoscale".

2) Lines 89-91: Since the authors are just showing simulated fields in Figs. 1a-c, the authors cannot simply write "in general agreement with observed values". The authors had better state **in what ways** these simulated fields are generally in agreement with observations.

- The simulated fields generally agree with the observed climatological fields. We have revised the description of Figure 1, as indicated in lines 90-94.

3) Line 99: To be more specific, replace "forcing" with "lateral boundary conditions".

- We replaced "forcing" with "lateral boundary conditions".

4) Lines 110-115: Since "q" is defined as the shortwave radiation in Eq. (2), it is not appropriate to use "q" in Eq. (3).

- We changed q to "Q" in Eq. (3).

5) Line 125: Add "dz" at the end of the left hand side of Eq. (6).

- Added "dz" to Eq. (6) on lhs.

6) Figure 1: Considering that the horizontal resolution of the ocean model is 2.5 km, the unit for the west-east distance in the horizontal axis and the north-south distance in the vertical axis cannot be km.

- The distance is fixed in figure 1. The distance was divided by 1000 by mistake. Please see attached "fig01.png"

7) Figures 2 and 3 caption: Why are the authors showing these six months with different intervals between them? Unless there is a clear reason, it is better to plot either odd or even

months. Also, replace "a)-c)" with "(a), (d), (g)", "d)-f)" with "(b), (e), (h)", and "g)-i)" with "(c), (f), (i)".

- These months were selected because they show the seasonal progression more clearly. The caption is fixed.

8) Figure 4: "January and March, 2016" in its figure caption is inconsistent with months in the title of each panel.

- It should be "March, June, and September, 2016"

9) Figures 5, 7, and 8: It is better to use black (or some other colors) lines instead of green lines for the Control experiment.

- We changed the green lines to black in Fig. 5, 7 and 8.

10) Figure 5a: Are the authors showing downward sea surface heat-flux anomalies as indicated in the figure caption or turbulent heat flux anomalies as written in the main text in Lines 200-207?

- We are showing turbulent heat flux anomalies. The caption is fixed.

11) Figure 6: Since figures should be labeled in the order of appearance, Fig. 6 should be labeled Fig. 2, because the figure is cited in Line 144 after Fig. 1.

- Figure.6 is now labeled Fig.2.

12) technical corrections
Line 70: Replace "dampen" with "damp".
- Corrected

Line 77: Add "by" after "detail".
- Corrected

Line 89: Replace "Figure." with "Fig.".
- Corrected

Line 94: Add "components" after "radiative".
- Corrected

Line 96: Replace "simulations" with "experiments"

– Corrected

Line 98: Replace "Eddy Kinetic Energy" with "eddy kinetic energy".
- Corrected

Line 98: Replace "ran" with "run", because the past participle of "run" is "run".
- Corrected

Line 104: Delete one of the "Li and Lee".
- Corrected

Line 110: Replace "flux" with "radiation".
- Corrected

Line 111: Add "and" before "q".
- Corrected

Line 114: Replace "heat flux" after "shortwave" and "longwave" with "radiation".
- Corrected

Line 115: Add "turbulent" before "latent".
- Corrected

Line 137: Replace "Li and Lee (2017)" with "(Li and Lee 2017)".
- Corrected

Line 169: Replace "in" with "of".
- Corrected

Line 178: Add "terms" after "residual".
- Corrected

Line 204: Replace "dampen" with "damp".
- Corrected

Figure 1 caption, Line 1: Delete one of the two "on".
- Corrected

Figure 7 caption, lines 2 and 3: Replace "experiment" with "experiments".
- Corrected

**Referee #2**

**Minor comments**

- Description of model domain. Lines 91–92: "Mesoscale anomalies are clearly visible in all fields (Fig.1)." Because this paper is focused on the ocean mesoscale dynamics, it is worth to add a snapshot of Eddy Kinetic Energy (EKE). I suggest replacing the SSS pattern (Fig. 1b) with an EKE pattern.

- We replaced SSS with EKE field in fig.1. We have revised the description of Figure 1, as indicated in lines 90-94.

- MLD pattern in model domain. In the model domain, which is a region in the Western Indian sector of the Southern Ocean, the deep MLD forms in the southern and eastern part of the domain (Fig. 1c). In fact, this deep MLD distribution is quite different from that in most other regions of the Southern Ocean, where the deep MLD forms on the northern flank of the ACC jets. More discussion/clarification is needed here.

- Thank you for bringing the confusion to our attention. It is important to note that Figure 1 displays a snapshot of MLD and exhibits several transient features. On the other hand, Figure 6f of Perlin et al. (2020) illustrates the mean and standard deviation of MLD, revealing deeper MLD formation in the northern flank of the ACC jets in the Southern Indian sector. We have clarified this in lines 90-94.

- Figure 6. Line 144: I am confused about the logic here. Should the authors show the correlation between SSTA and MLD, instead of SSTA and MLDA, in Fig. 6? In this way, the authors can contrast the contributions of SST with and without mesoscale anomalies to the MLD variability.

- We aim to investigate the processes responsible for the mesoscale MLD variability. From a large-scale perspective, there is a positive correlation between SST and MLD, indicating that cooler (warmer) SST leads to deeper (shallower) MLD. However, from a mesoscale perspective, there is low correlation between SST and MLD, suggesting that their relationship becomes more complex for mesoscale anomalies.

    - Figure 8. Can the authors comment on why the RMS MLD anomalies in the two sensitivity experiments show a similar response in time?

- The reviewer may be wondering why mesoscale heat flux and momentum flux drive similar MLD responses over time. We believe that seasonal variations are a significant factor in the RMS MLD anomalies. Specifically, during the summer season, the MLD is shallower, and atmospheric forcing is typically more influential. Conversely, during winter, the MLD is deeper, and atmospheric forcing is generally less critical.

    - Figure 9. The signals of two sensitivity experiments in the domain average are small and not very clear. I suggest to conduct the same calculation, but only averaged for the areas where the wintertime MLDs in the control experiment are deeper than a certain threshold,

i.e. ≥400 m. I expect that the signals would become clearer.

- We set the threshold to be 200 m and the signals are indeed stronger but not necessarily clearer. The time series still show the same conclusion (see fig09v2.png).

- Figure 3. There are some 'white spots' close to the surface in Fig. 3e, f. This issue can be solved by modifying the color bar.

-We modified the color bar and the white spots are gone.

---

## Author Response (AR2)

Reply to suggestions for revisions

General comments:
The authors have incorporated my comments from the first round of review and the manuscript has improved. In particular, it is nice that the term "eddy" is more precisely defined in the revised manuscript. However, there are some obvious errors in their figures and I have very minor comments. Thus, I recommend minor revisions prior to publication of this manuscript.

Minor comments:
(Line numbers refer to those in the manuscript with tracked changes.)
1) Line 6: Replace "mixed layer depth" with "MLD".
2) Line 9: It is better to use "mesoscale ocean current anomalies" instead of "ocean eddies" here.
3) Lines 95-97: I agree with the authors that the new names for sensitivity experiments better represent the experiments. I recommend the authors to also state whether wind stress is calculated relative to surface current or not.
4) Line 108: "g" should also be defined.
5) Lines 147-148: It seems that the sentence "We agree that it is important to clarify that in our analysis, "eddy" refers not only to coherent vortices but also to all mesoscale processes that the filter can smooth out." is a part of the authors' response to my comment in the first round of review an should be deleted.
6) Line 254: Replace "domain" with "domains".
7) Figure 1: Replace "sea surface salinity" with "eddy kinetic energy" and "eddy kinetic energy" with "mixed layer depth (MLD)".
8) Figure 3: Replace "a)-c)" with "(a), (d), (g)", "d)-f)" with "(b), (e), (h)", and "g)-i)" with "(c), (f), (i)".
9) Figures 6-9: The authors forgot to update figure legends after changing the names of sensitivity experiments.

1) Line 6: We replaced "mixed layer depth" with "MLD".
2) Line 9: We agree it is better to be specific about the fact that we are addressing the advection by mesoscale ocean current, instead of ocean eddies in general.
3) Line 95-97: Wind stress is calculated from the model friction velocity and 10-m wind directional components (Perlin et al, 2020), and is not calculated relative to surface current.
4) Line 108: "g" represents the gravitational acceleration.
5) Lines 147 - 148: We deleted that part of the sentence.
6) Line 254: We replaced "domain" with "domains".

7) Figure 1: We fixed the caption.
8) Figure 3: We replaced "a)-c)" with "(a), (d), (g)", "d)-f)" with "(b), (e), (h)", and "g)-i)" with "(c), (f), (i)".
9) Figures 6-9: We updated the figure legends.